# Ensemble forecasts of COVID-19 activity to support Australia's pandemic response: 2020–22

Robert Moss[1]*, Ruarai J. Tobin[1], Mitchell O'Hara-Wild[2], Adeshina I. Adekunle[3], Dennis Liu[4], Tobin South[4], Dylan J. Morris[4], Gerard E. Ryan[1,5], Tianxiao Hao[1,5], Aarathy Babu[5], Katharine L. Senior[1,5], James G. Wood[6], Nick Golding[5,7], Joshua V. Ross[8,9], Peter Dawson[10], Rob J. Hyndman[2], David J. Price[1,11☉], James M. McCaw[1,12☉], Freya M. Shearer[1,5☉]

1 Melbourne School of Population and Global Health, The University of Melbourne, Melbourne, Victoria, Australia, 2 Department of Econometrics and Business Statistics, Monash University, Melbourne, Victoria, Australia, 3 Defence Science and Technology Group, Melbourne, Victoria, Australia, 4 School of Computer and Mathematical Sciences, The University of Adelaide, Adelaide, South Australia, Australia, 5 The Kids Research Institute Australia, Perth, Western Australia, Australia, 6 School of Population Health, University of New South Wales, Sydney, New South Wales, Australia, 7 School of Physics, Mathematics and Computing, University of Western Australia, Perth, Western Australia, Australia, 8 Department for Health and Wellbeing, Government of South Australia, Adelaide, South Australia, Australia, 9 South Australian Health and Medical Research Institute, Adelaide, South Australia, Australia, 10 Sensors and Effectors Division, Defence Science and Technology Group, Department of Defence, Melbourne, Victoria, Australia, 11 Department of Infectious Diseases, The Peter Doherty Institute for Infection and Immunity, The Royal Melbourne Hospital and The University of Melbourne, Melbourne, Victoria, Australia, 12 School of Mathematics and Statistics, The University of Melbourne, Melbourne, Victoria, Australia

☉ These authors contributed equally to this work.
* rgmoss@unimelb.edu.au

## Abstract

During the COVID-19 pandemic, many countries used real-time data analyses, predictive modelling, and COVID-19 case forecasts, to incorporate emerging evidence into their decisions. In Australia, national and jurisdictional public health responses were informed by weekly ensemble forecasts of daily COVID-19 case counts for each of Australia's eight states and territories, produced by a consortium of researchers under contract with the Australian Government. As members of this consortium, who produced these forecasts at each week, we now retrospectively evaluate approximately 100,000 predictions for daily case counts 1–28 days into the future, generated between July 2020 and December 2022, and report here (a) how the ensemble forecasts supported public health responses; (b) how well the ensemble forecast performed, relative to the forecasts produced by each contributing team; and (c) how we refined our reporting and visualisations to ensure that outputs were interpreted appropriately. Similar to COVID-19 forecasting studies in other countries, we found that the ensemble forecast consistently out-performed the individual model forecasts, and that performance was lowest when there were rapid changes in the epidemiology, such as periods around epidemic peaks. Our consortium's internal peer-review process allowed us to explain how features of each ensemble forecast related to the

**Data availability statement:** The analysis code, summary outputs, figures, and supplementary materials are provided in a public git repository: https://gitlab.unimelb.edu.au/rgmoss/analysis-australian-covid19-ensemble-forecasts-results Note that this repository does not include the input cases files and output ensemble forecasts (approximately 2.3 Gb). These are provided in a separate data repository: https://doi.org/10.26188/29434298.

**Funding:** This work was directly funded by the Australian Government Department of Health and Aged Care, as urgent public health action to support Australia's COVID-19 pandemic response. Additional support was provided by the National Health and Medical Research Council of Australia through its Centres of Research Excellence (SPECTRUM, GNT1170960) and Investigator Grant Schemes (FMS, Emerging Leader Fellowship 2021/GNT2010051). The funders did not play any role in the study design, data analysis, decision to publish, or preparation of the manuscript.

**Competing interests:** The authors have declared that no competing interests exist.

design of the individual models, and this helped enable public health stakeholders to interpret the forecasts appropriately. Ultimately, our forecasts provided information that supported public health responses during periods of different policy goals, and over a wide range of epidemic scenarios.

## Author summary

In response to the emergence of COVID-19, governments around the world implemented a range of measures to reduce infections and deaths. In many countries, public health responses were guided by data analyses and by predictive forecasts of future cases, hospitalisations, and deaths. The authors were part of a consortium that was contracted by the Australian Government to produce forecasts of daily COVID-19 case counts for each of Australia's eight states and territories from July 2020 to December 2023. Similar to COVID-19 forecasting studies in other countries, we found that combining forecasts from multiple models into an ensemble forecast consistently out-performed the individual models, and that performance was lowest when there were rapid changes in the epidemiology. We show here how these forecasts supported public health responses during periods of different policy goals, and over a wide range of epidemic scenarios.

## Introduction

The COVID-19 pandemic resulted in major social disruptions around the world [1,2], as governments implemented a range of measures to curb infections and limit deaths in the face of a novel virus [3–5]. Initial decisions were guided by public health response plans, and as the pandemic unfolded governments increasingly made use of modelling to incorporate emerging evidence into their decisions [6–11]. Australia's national and jurisdictional public health responses were informed by the results of real-time analyses that were presented to key government advisory committees in weekly situation reports, starting 4th April 2020 [12,13]. Between July 2020 and December 2023, these reports included ensemble forecasts of daily COVID-19 case counts for each of Australia's eight states and territories. A timeline of key events during this period is presented in Table 1.

Beginning on 1st February 2020, the Australian government progressively closed borders to countries with established epidemics, culminating in almost complete cessation of travel from 20th March 2020, and imposed mandatory 14-day hotel quarantine on overseas arrivals from 16th March 2020. These measures were effective in limiting community exposure from infected international arrivals. Australia's initial COVID-19 response focused on preventing local transmission [14] and, despite a steady influx of imported cases, local elimination was achieved for prolonged periods throughout 2020 and 2021, punctuated by several waves of local transmission. Over the 8 months from 1st October 2020–1st June 2021 there was a total of 874 reported

**Table 1. A timeline of key events prior to, and during, the study period (July 2020 to December 2022, inclusive).**

| Year | Month | Event |
|------|-------|-------|
| 2020 | Jan | First cases imported |
|      |     | National goal of preventing local transmission is announced |
|      | Mar | Limited local transmission occurs nationally |
|      |     | Hotel quarantine established for returning residents |
|      | Jun | Sustained local transmission occurs in Victoria |
|      | Oct | Local transmission is suppressed in Victoria |
| 2021 | Feb | Vaccine roll-out for priority and high-risk individuals begins |
|      | May | Vaccine roll-out for all adults begins |
|      | Jul | Delta variant becomes established in NSW (and subsequently in other jurisdictions) |
|      | Aug | National reopening plan targets are announced |
|      | Nov | NSW is the first jurisdiction to remove quarantine requirements for returning residents |
|      | Dec | Omicron BA.1 becomes established in all jurisdictions except WA |
| 2022 | Jan | Omicron BA.1 activity peaks in affected jurisdictions |
|      |     | Vaccine roll-out for children aged 5–11 begins |
|      |     | Booster doses recommended for at-risk individuals |
|      | Feb | Borders opened to all vaccinated travellers except in WA |
|      | Mar | Omicron BA.2 and BA.5 become established |
|      |     | WA is the last jurisdiction to remove quarantine requirements |
|      | Jul | Omicron BA.5 activity peaks |
|      | Sep | Multiple variants become established |
|      | Dec | Multiple variants activity peaks |

cases, of which 345 were in the state of Victoria. Australia's COVID-19 vaccination program began in February 2021, with the aim of pivoting from preventing local transmission to "reopening" the country (i.e., easing non-pharmaceutical interventions). Vaccination targets and reopening plans were informed by modelling studies [15–17]. Beginning on 1st November 2021, Australian jurisdictions progressively relaxed quarantine requirements for returning fully-vaccinated Australian residents and citizens, and subsequently for international travellers, with quarantine requirements fully removed as of 3rd March 2022. Over the 2022 calendar year Australia experienced substantial local transmission of multiple COVID-19 variants, with major waves of Omicron BA.1, BA.2, and BA.5, consistent with global epidemiology [18]. Many of these policy decisions were directly informed by our situation reports and the ensemble forecasts contained within. These forecasts also served as an input for COVID-19 hospital bed occupancy forecasts (December 2021 to December 2023) that further supported Australia's public health responses [19].

Ensemble forecasts combine forecasts from multiple models into a single forecast. This approach was first proposed by Bates and Granger [20] in 1969, who showed that combining two sets of forecasts of airline passenger data yielded smaller errors than either of the original forecasts. Ensemble forecasts have been the standard approach for numerical weather prediction and climate since the 1990s [21]. Empirical evidence and extensive simulations show that the simple average of candidate forecasts is an effective and robust combination method that often outperforms more complicated methods; this has become known as the "forecast combination puzzle" [22,23]. In the context of infectious diseases, ensemble forecasting has been an active area of research of the past decade, with applications to diseases such dengue [24] and, most prominently, the United States Centers for Disease Control and Prevention's "FluSight" program that began in 2013 [25–27]. During the COVID-19 pandemic, the United States of America and Europe both established public COVID-19 forecast hubs with open submission policies, and combined submitted forecasts into ensemble forecasts [9,10].

   

Consistent with the forecast combination puzzle, both hubs reported that their ensembles were consistently among the best-performing forecasts, and that weighting individual models by past performance did not improve the ensemble.

Here we present an analysis of the ensemble forecasts of COVID-19 cases that were produced by the authors, under contract with, and reported weekly to, the Australian Government. We demonstrate how the ensemble consistently out-performed the individual models that were included in the ensemble, and how these forecasts supported public health responses over the 2020–2022 calendar years. Because we were directly funded by the Australian Government to support Australia's public health responses, clear and effective communication of forecast outputs to stakeholders was of paramount concern, and we reflect on how we refined our reporting and visualisation over the study period. Relative to the USA and European COVID-19 hubs, distinguishing features of this work include our internal peer-review process, our detailed knowledge concerning the methods and implementation of each contributing model (due in part to our much smaller scale), and our use of daily case counts which, when compared to using weekly counts, adds complexity regarding test-seeking and diagnosis timelines. We conclude by highlighting the relative strengths and weaknesses of our ensemble approach, relative to the USA and European COVID-19 hubs.

## Materials and methods

### Ethics statement

The study was undertaken as urgent public health action to support Australia's COVID-19 pandemic response. The study obtained data under the National Health Security Agreement for the purposes of national communicable disease surveillance. Contractual obligations established strict data protection protocols, as agreed between the University of Melbourne and sub-contractors and the Australian Government Department of Health and Aged Care. Oversight and approval for use in supporting Australia's pandemic response, and for publication, were provided by the data custodians (represented by the Communicable Diseases Network of Australia, CDNA). The ethics of the use of these data for these purposes, including publication, was agreed by the Department of Health and Aged Care with CDNA. Consent was not obtained, these data were anonymised at the source and were not reidentifiable. All methods were carried out in accordance with the relevant guidelines and regulations.

### COVID-19 case data

For target data, we used de-identified line lists of reported cases for each Australian state and territory, extracted at each week of the study period from the National Notifiable Disease Surveillance System (NNDSS). The data included the jurisdiction (state or territory), the date of symptom onset (where available), the date when the case notification was received by the jurisdictional health department, and whether the infection was epidemiologically deemed to be acquired locally or overseas. We imputed missing symptom onset dates and estimated reporting delays using a time-varying delay distribution to characterise the duration between symptom onset and case notification [28]. Symptom onset dates were reported for 82% of cases in 2020. As Australia's response pivoted to "reopening" the country, and daily case counts greatly increased, symptom onset reporting decreased to 48% of cases in 2021, and 38% in 2022. Nationally aggregated daily case counts are shown in Fig 1.

### Forecasting teams and models

Four modelling teams contributed forecasts for inclusion in the ensemble, using different types of model (see Table 2). This included two stochastic compartmental models, developed by the Defence Science and Technology Group (DST) and the University of Melbourne (UoM), which incorporated different assumptions about effective reproduction numbers and population immunity; a branching process model developed by the University of Adelaide (UoA); and an autoregressive model developed by Monash University (MU). We began with three teams contributing to the ensemble (MU, UoA,

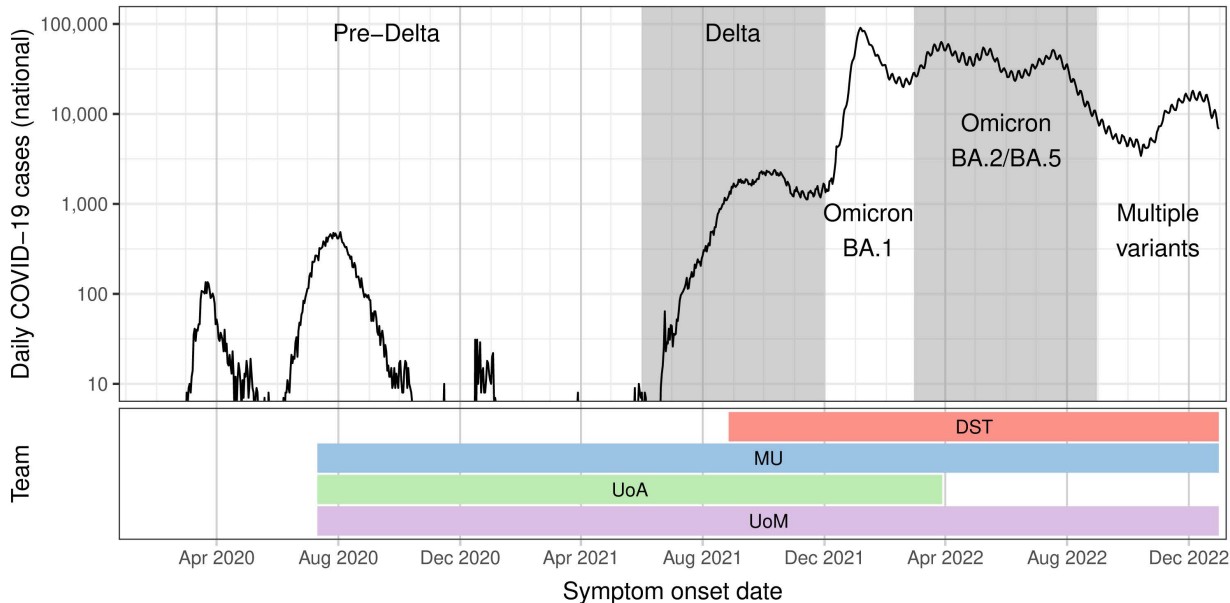

**Fig 1. National daily confirmed COVID-19 cases and participating teams.** Top: Australian daily confirmed COVID-19 cases for the 2020–2022 calendar years. Dominant strain(s) are identified by alternating backgrounds (white and shaded), as determined by epidemiological assessment in situation reports [13]. Bottom: The periods over which each team contributed to the ensemble forecast. Team names and modelling approaches are listed in Table 2.

**Table 2. The teams that contributed to the ensemble forecasts over the study period.**

| Institute | Methodology |
| --- | --- |
| Defence Science Technology (DST) | Stochastic SEIR-type compartmental model |
| Monash University (MU) | Global autoregression model |
| University of Adelaide (UoA) | Branching process model |
| University of Melbourne (UoM) | Stochastic SEIR-type compartmental model |

and UoM) and the DST team began participating in August 2021. After a four-month period where all four teams contributed to the ensemble forecast (2021-11-25 to 2022-03-22, inclusive), the UoA team stopped participating due to limited staff availability. Throughout this manuscript we will refer to the models, and the forecasts that they produced, using the acronyms listed in Table 2 and used in this paragraph.

The UoM team began contributing national forecasts in April 2020 [29], using a model adapted from a long-running seasonal influenza forecasting project [30–33], while the other teams created new models in response to the emergence of local COVID-19 transmission. All models were subject to ongoing development throughout the study, and these developments are documented in our weekly situation reports [13]. When new model features were introduced, retrospective forecasts were validated against previously-reported data before the new model iteration was considered for inclusion in the ensemble.

Teams were able to use other data sources, in addition to the Australian COVID-19 case data. The MU team used global COVID-19 case data from Johns Hopkins [34] to inform their global autoregression model. The DST and UoA teams made use of Google mobility data to inform transmission estimates, the DST and UoM teams made use of behavioural survey data to inform case ascertainment [35], and the DST, UoA, and UoM teams made use of vaccination data from the Australian Immunisation Registry to inform vaccine coverage in their models. Where multiple teams used a

common data source, they did not necessarily do so in an identical manner. For example, the DST and UoM teams made different assumptions about how much of the variation in case numbers reported by day-of-week was attributed to transmission versus case ascertainment.

## Inclusion/exclusion criteria

If there were known, or strongly suspected, data quality issues for any jurisdiction that could not be resolved with post-processing, we did not include ensemble forecasts for those jurisdictions in the weekly situation report. Models were also excluded when the forecasts produced for that week were assessed to be inappropriately sensitive to trends in the most recent case counts, which were known to be subject to time-varying reporting delays [28]. This decision-making process was facilitated by direct communication with data custodians in the relevant jurisdictions and national public health committees, which enabled rapid awareness of data quality issues.

Each week, the modelling teams generated forecasts from the provided data extracts and shared the results for peer review by the broader analysis team (comprising all modelling teams and other analysts, which we will refer to as the "consortium" hereafter) in a private online discussion board (in a Slack workspace). Peer review findings and retrospective performance results for each model were then tabled for discussion in weekly online consortium meetings, where we collectively reached agreement on which models, and model iterations, to include in the ensemble forecast for each jurisdiction that week (see Fig 2). This approach of ongoing discussion and reaching a collective consensus has been shown to improve the performance of human predictions in domains such as geopolitics [36].

Models were excluded when their outputs differed dramatically from the consortium's collective expectations and the current epidemiological assessment, or when a technical issue was identified by the modelling team or by the peer review process. Importantly, each model was evaluated against the most recent data rather than competing against the other models (e.g., via benchmarks such as skill scores), with the purpose of deciding whether the model outputs were likely to be a reliable basis for decision-making [37]. In other words, because we did not believe that any participating model truly captured the real-world system, we therefore did not use performance metrics to undertake model selection. Rather, teams were able to use these metrics to inform model development. The outcomes of these collective decisions are captured in publicly available reports [13].

## Ensemble forecast construction

Model forecasts were submitted in the form of 2000 simulated trajectories of daily case counts over the 28-day forecast horizon for each of the eight jurisdictions. We generated equal-weight ensemble forecasts for each jurisdiction by sampling trajectories from each team's forecasts for that jurisdiction.

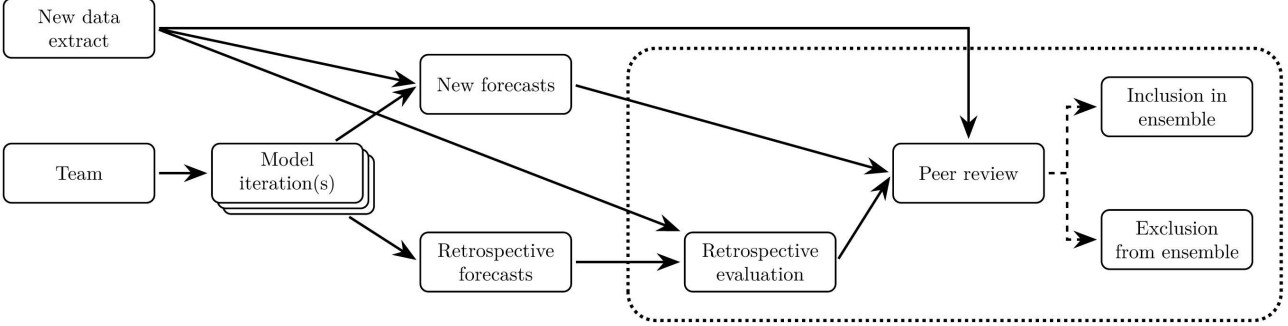

**Fig 2. A flowchart of the weekly peer review and inclusion/exclusion decision process.** Shown here is an example for the model iteration(s) provided by a single modelling team in a given week. The dotted box indicates activities and decisions made by the entire consortium.

When a team provided forecasts from multiple model iterations, we treated these forecasts as arising from a single model with a discrete hyper-parameter (i.e., having a different value for each model iteration), and averaged over the iterations when including the team's forecasts in the ensemble. We believe that this was the most sensible approach to take, particularly when our primary interests were to evaluate how well each model performed relative to the other models, and relative to the ensemble. The modifications made to each model throughout the study period were much smaller than the methodological and structural differences between the models. Accordingly, the addition or removal of one model iteration impacted the relative weights of all model iterations from that team, but did not affect the weights allocated to other teams.

The use of equal weights for each team was strongly motivated by the risk of sub-optimal forecasts supporting a poor decision/recommendation. An ensemble consistently performs better than a single model, and there is a large body of evidence in the forecasting literature that it is very hard to out-perform an equal-weight ("simple average") ensemble; this is known as the "forecast combination puzzle" [22,23]. This reliability of a "simple average" ensemble was more valuable for our purposes than any potential (and likely very small) gain in performance from using a more complex weighting scheme.

For a given week $w$, each team $t \in T$ provided forecasts $F_t(w)$ for $M$ model iterations $\{F_t^1(w), \ldots, F_t^M(w)\}$. Teams provided $k = 2000$ sample trajectories $\phi_t^m(w)$ for each model iteration, and we accepted a subset $M_t(w)$ of these iterations for inclusion in the ensemble. Weights $\sigma_t^m$ were distributed equally across the $N(w)$ teams with at least one model iteration included in the ensemble forecast:

$$\sigma_t^m(w) = \frac{\mathbf{1}_{M_t}[F_t^m(w)]}{N(w) \cdot \max\left(1, |M_t(w)|\right)}.$$

The ensemble forecast distribution $F_{\text{ens}}(w)$ was defined to be the weighted sum of model iteration forecasts $F_t^m(w)$ :

$$F_{\text{ens}}(w) = \sum_{t,m} \sigma_t^m(w) \cdot F_t^m(w).$$

As described above, we held weekly meetings in which we reviewed individual model forecasts and the ensemble forecast prior to including these forecasts in our weekly situation reports to government. Forecast performance was evaluated visually (using daily credible intervals) against previous weeks' forecasts and the most recent data and quantitatively, as we explain in the following sections.

### Forecast performance

We quantified forecast performance using the Continuous Ranked Probability Score (CRPS). CRPS is a generalisation of the Mean Absolute Error (MAE) to probabilistic forecasts, and evaluates the entire forecast distribution $F$ against the observed data $y$:

$$\text{CRPS}(F, y) = \int_{\mathbb{R}} \left[F(x) - \mathcal{H}(x - y)\right]^2 \, dx.$$

where $\mathcal{H}$ denotes the Heaviside step function.

It is interpreted on the natural scale of the data $y$, with a minimum value of zero indicating a perfect point forecast. CRPS is a proper scoring rule, which means that its expected score is minimised when the forecast distribution matches the distribution of the observations, and is the most commonly used scoring rule in the evaluation of probabilistic forecasts. When evaluated for log-transformed distributions against log-transformed observed values $\log(y + 1)$, it can be interpreted as a probabilistic version of the relative error [38].

We used CRPS skill scores to compare the relative performance of each team's forecasts $F_t$ against that of the ensemble $F_{ens}$, for the observed data $y$:

$$\text{Skill}(F_t, y) = \frac{\text{CRPS}(F_{ens}, y) - \text{CRPS}(F_t, y)}{\text{CRPS}(F_{ens}, y)}.$$

The maximum skill score is 1 (obtained for a perfect point forecast $F_t$). Positive skill scores indicate that the forecast $F_t$ has out-performed the reference forecast (in this case, the ensemble forecast $F_{ens}$ ) while negative skill scores indicate that the forecast $F_t$ has performed worse than the reference forecast.

### Forecast calibration

Forecasts were visualised as median-centred credible intervals for daily case counts in each jurisdiction. For example, the 50% credible interval for a forecast distribution $F$ was calculated as the inter-quartile interval [0.25, 0.75], and the 95% credible interval as the inter-quartile interval [0.025, 0.975].

Credible intervals represent the plausibility that the observed data $y$ will fall within the interval range. The coverage of a given $X$% credible interval is the observed proportion of the data $y$ that fall within this credible interval. When evaluated over a range of credible intervals, this quantifies how well calibrated the forecasts $F$ are, with respect to the observed data $y$.

A set of forecasts $F = \{F_1, F_2, \ldots\}$ is probabilistically calibrated with respect to observed data $y = \{y_1, y_2, \ldots\}$ if there is statistical consistency between these two sets, such that $X$% of the data $y$ fall within the $X$% credible interval of the forecasts $F$ [39].

### Forecast bias

We also evaluated forecast bias $B(F, y)$ as per Funk et al. [40]:

$$B(F, y) = 1 - \left[ P(F \leq y) + P(F \leq y - 1) \right].$$

An unbiased forecast would assign equal probability mass to values above and below the observed data $y$ ($B(F, y) = 0$), while a completely-biased model would assign all probability mass above $y$ ($B(F, y) = 1$) or below $y$ ($B(F, y) = -1$).

## Results

We begin with an overview of the model inclusion/exclusion outcomes for each ensemble forecast. We then present an overview of the ensemble forecast performance, and demonstrate that the ensemble forecasts were more reliable than the forecasts generated by each individual model. We conclude this section by detailing how the ensemble forecasts supported public health activities during each phase of the study period (defined by predominant circulation of different COVID-19 variants, as illustrated in Fig 1) and highlight ways in which we improved forecast visualisations and reporting over the study period.

### Models included in each ensemble forecast

As shown in Table 3, all models were regularly included in the ensemble forecasts. The DST and UoM teams often contributed forecasts generated from multiple model iterations, and when models were excluded from the ensemble it was most often *a single model iteration* from either team. The most common reason for exclusion was for a model to exhibit large and systematic biases in the forecasts produced in previous weeks (i.e., retrospective posterior predictive checks). In several instances the UoA team encountered computational issues and were unable to provide forecasts in a timely

**Table 3. A summary of models included in the ensemble forecasts.** The percentage of ensemble forecasts for which each model was included, and the number of models included in each ensemble forecast (minimum, maximum, and mean), reported separately for each period in which particular strain(s) dominated. Note that models could be excluded from the ensemble for various reasons, from being assessed as inappropriately sensitive to trends in the most recent data, or otherwise producing suspect predictions, to being unavailable due to technical issues and/or delays in model development (see "Inclusion/exclusion criteria").

| Strain | Forecasts | DST | MU | UoA | UoM | # Models |
|---|---|---|---|---|---|---|
| Pre-Delta | 302 | — | 100.0% | 93.7% | 100.0% | 2–3 (2.99) |
| Delta | 183 | 4.4% | 100.0% | 100.0% | 100.0% | 3–4 (3.04) |
| Omicron BA.1 | 94 | 84.0% | 100.0% | 57.4% | 100.0% | 2–4 (3.41) |
| Omicron BA.2/BA.5 | 208 | 96.2% | 96.2% | 7.7% | 100.0% | 2–4 (3.00) |
| Multiple variants | 89 | 100.0% | 100.0% | — | 98.9% | 2–3 (2.99) |
| | 876 | 376 | 868 | 536 | 875 | 2–4 (3.03) |

manner. Due to staff leave, some teams did not provide forecasts over the Christmas 2021 holiday period (during "Omicron BA.1"). At each week of the study period, we accepted forecasts from multiple teams (with a minimum of two teams) and always produced an ensemble forecast, since an ensemble is expected to out-perform any individual model.

## Summary of forecast performance

Ensemble forecasts for the months of March, April, and May 2022 are shown in Fig 3 for four of the eight Australian jurisdictions (forecasts for all jurisdictions over the entire study period are shown in Figures A–E in S1 Text). These example forecasts demonstrate key features that were exhibited by the ensemble forecasts over the entire study period (2020–2022). In brief, the ensemble was generally in good agreement with the data. Consistent with findings from COVID-19 forecasting studies in other countries [6–11,41], it was an ongoing challenge to predict the timing of change points. The forecasts exhibited a tendency to overshoot peaks, and to undershoot the onset and ending of individual waves. In particular, the forecasts tended to predict substantially larger peaks than were ultimately observed. However, as shown in the 4-week forecasts in Fig 3, the observed peaks still fell within the forecast credible intervals. The extremely broad intervals for the Victorian forecast for 15 March 2022 arise from a single model, which was assessed as highly uncertain but consistent with the data, and included in the ensemble that week.

Ensemble forecast coverage (shown in Fig 4) was reasonable across most pandemic waves. The best coverage was observed during the "Pre-Delta" and "Delta" periods where, for example, almost exactly 50% of the ensemble 50% credible intervals included the true value. During this time, Australian jurisdictions responded rapidly to local disease activity in pursuit of strong suppression [14], and while the precise impact of these interventions was challenging to predict, there was only limited local COVID-19 transmission and case counts remained low. Australia then pivoted from pursuing strong suppression to reopening and accepting substantial levels of local COVID-19 transmission. In the subsequent "Omicron BA.1" period, the models failed to anticipate the massive surge and rapid decline in daily case counts; forecasts were overly confident and exhibited a tendency to first undershoot and then overshoot the true case counts. Forecast coverage improved markedly for the "Omicron BA.2/BA.5" and "Multiple variants" periods, with a slight tendency towards being overly uncertain, and was almost as good as during the "Pre-Delta" and "Delta" periods. This was likely due to a combination of having quantitative data about trends in test-seeking behaviour [35] (to inform the mechanistic models) and the slower rates of growth and decline in cases, relative to the "Omicron BA.1" period. For periods where the individual models exhibited marked differences in forecast coverage, the ensemble forecast coverage was similar to, if not better than, the coverage of the best individual model.

As might be expected, observations that lay further into the future were harder to predict. Forecast performance was highest for short lead times, and steadily decreased as the lead time increased. This was true not only on aggregate, but also for the majority of individual observations. As shown in Table 4, for a given observation the forecast performance

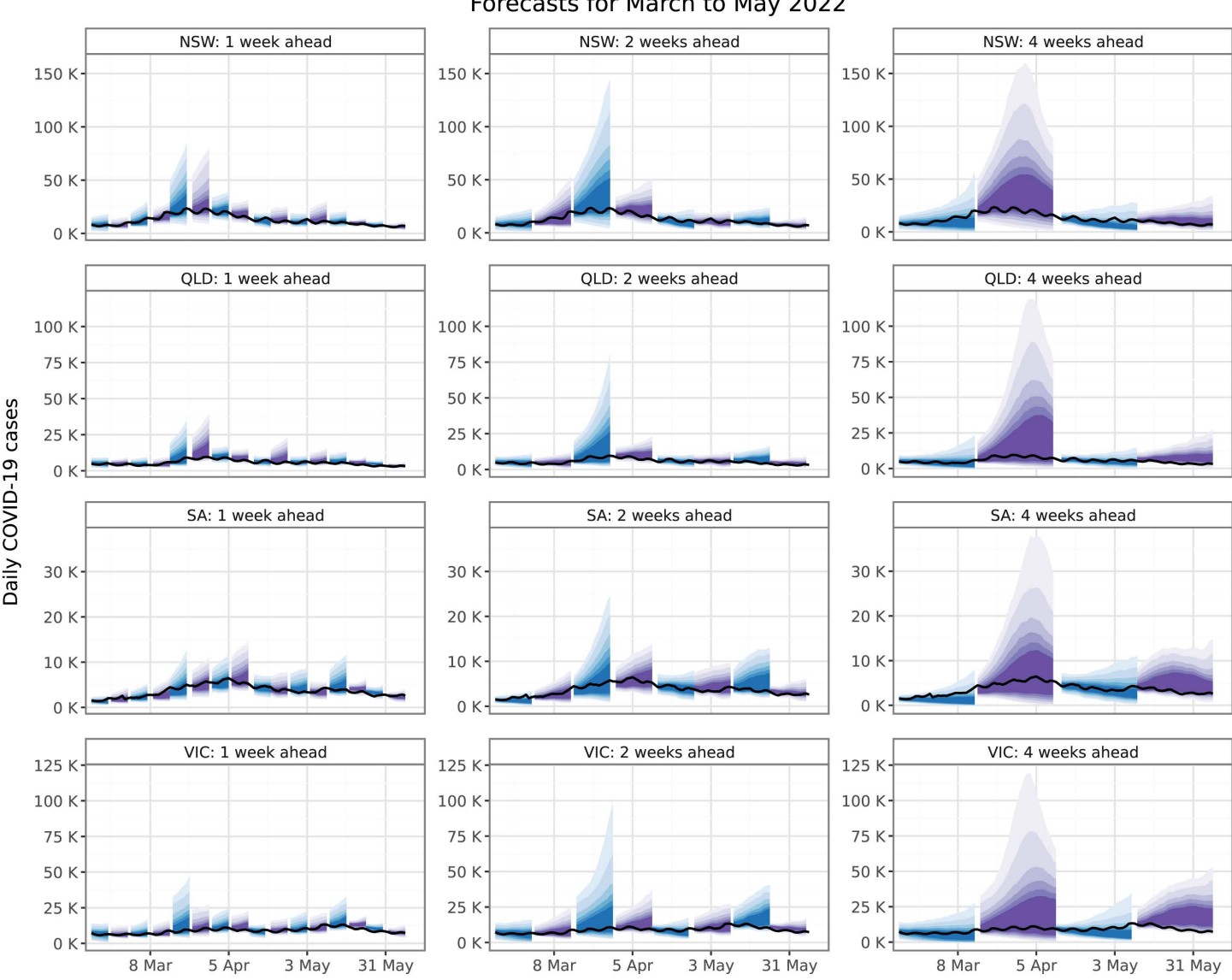

**Fig 3. Ensemble forecasts over the months of March, April, and May 2022.** Forecasts are shown for four Australian jurisdictions: New South Wales (NSW); Queensland (QLD); South Australia (SA); and Victoria (VIC). The left columns shows 1-week forecasts for each ensemble forecast (1–7 days ahead), the middle column shows 2-week forecasts for every second ensemble forecast (1–14 days ahead), and the right column shows 4-week forecasts for every fourth ensemble forecast (1–28 days ahead). Shaded regions illustrate the 50%, 60%, 70%, 80%, 90%, and 95% credible intervals, and black lines depict the data as reported at the end of the study period.

consistently improved from the initial prediction (when the observation was 22–28 days ahead) to the final prediction (when the observation was 1–7 days ahead).

Forecasts exhibited substantial biases around epidemic peaks. This is illustrated in Fig 5, which shows the bias for each model and the ensemble for forecasts generated in the 4 weeks leading up to, and the 4 weeks after, each observed peak. Forecasts generated 2–4 weeks prior to observed peaks undershot the data (negative bias) more often than they overshot the data (positive bias). In contrast, forecasts generated from 1 week prior to 1 week after the observed peaks

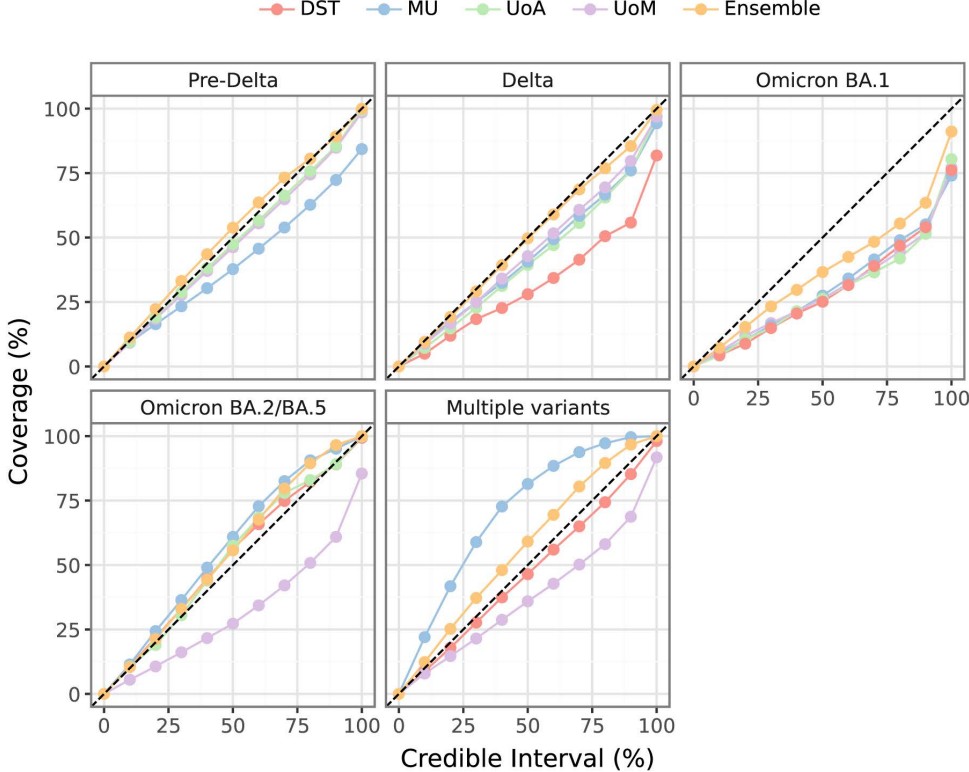

**Fig 4. Observed forecast credible interval coverage.** Observed coverage of the forecast credible intervals for each individual model, and for the ensemble, with respect to the ground truth case counts as reported after the end of the study period. Results are reported with respect to each dominating strain (refer to Fig 1). Dashed lines indicate perfect coverage. Points below the dashed lines indicate overly certain forecasts, points above the dashed lines indicate overly uncertain forecasts.

**Table 4. Ensemble forecast (log-transformed) CRPS values. These values can be interpreted as a probabilistic version of relative error. Mean values are shown for lead times of 1–7 days ("1 week"), 8–14 days ("2 weeks"), 15–21 days ("3 weeks"), and 22–28 days ("4 weeks") in each jurisdiction, and across all jurisdictions (bottom row). The forecasts performed best at the shortest lead times (i.e., when CRPS values were smallest) and performance decreased in proportion to lead time (i.e., CRPS values increased).**

| Jurisdiction | 1 week | 2 weeks | 3 weeks | 4 weeks |
|---|---|---|---|---|
| ACT | 0.153 | 0.263 | 0.391 | 0.503 |
| NSW | 0.237 | 0.387 | 0.512 | 0.644 |
| NT | 0.130 | 0.205 | 0.284 | 0.369 |
| QLD | 0.253 | 0.389 | 0.522 | 0.655 |
| SA | 0.201 | 0.287 | 0.369 | 0.479 |
| TAS | 0.116 | 0.213 | 0.329 | 0.429 |
| VIC | 0.229 | 0.361 | 0.480 | 0.583 |
| WA | 0.150 | 0.197 | 0.271 | 0.339 |
| National | 0.184 | 0.288 | 0.395 | 0.501 |

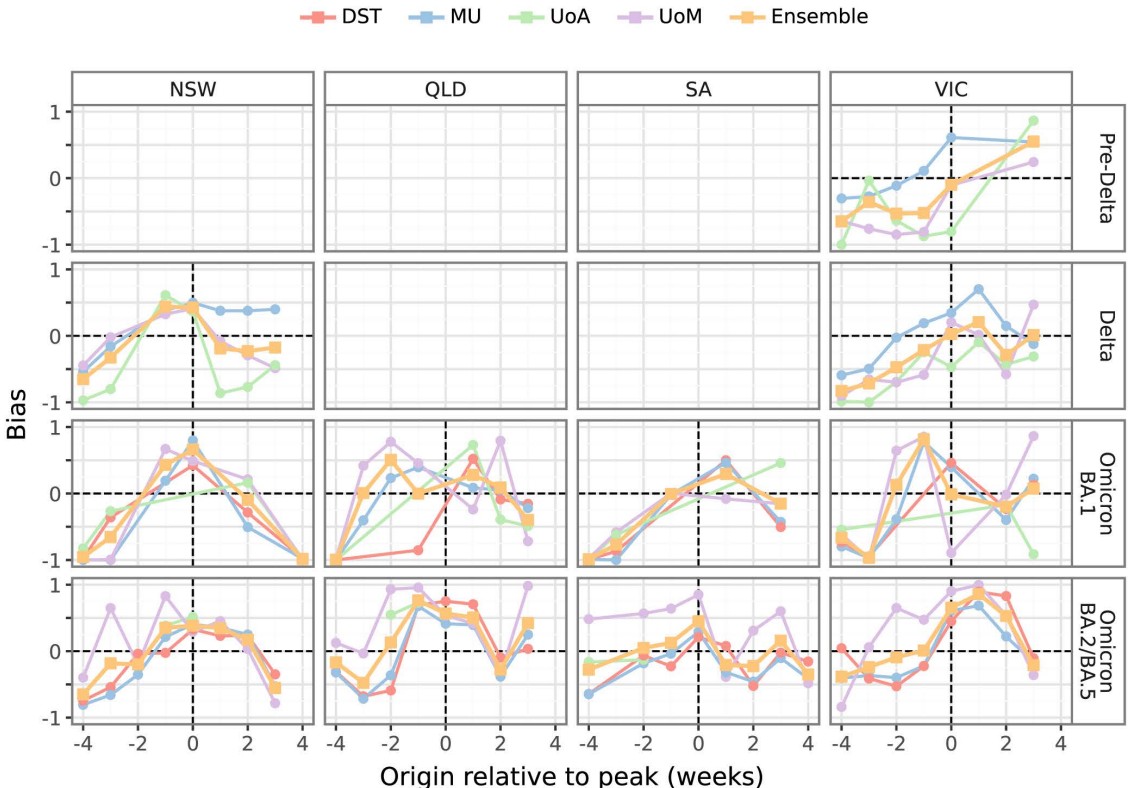

**Fig 5. Forecast bias around peaks.** Forecast bias for each model and the ensemble, shown for forecast origins within 4 weeks either side of the largest peak (minimum 200 cases) observed in four jurisdictions (columns) for each dominant strain (rows). Positive bias indicates a tendency to overshoot the data, negative bias indicates a tendency to undershoot the data. The "Multiple variants" period is not included in the figure, because case counts were generally flat and the observed peaks occurred towards the end of the study period. See Figure F in S1 Text for forecast bias around these peaks over all eight jurisdictions.

had a tendency to overshoot the data (positive bias), which highlights the challenge of identifying peaks in real-time when surveillance data are subject to ascertainment biases and reporting delays [12].

No model consistently produced the least biased forecasts for weeks preceding or following the observed peaks. When there was a wide spread in bias, the ensemble forecast was often less biased than most, if not all, of the individual model forecasts. Similar trends are evident in individual model and ensemble forecast performance (see Figures G and H in S1 Text); no model consistently produced the most accurate forecasts for weeks preceding or following the observed peaks, and the ensemble tended to mostly out-perform individual models.

While the ensemble forecasts struggled to accurately predict peak sizes and timing (evident in the measure of forecast bias, discussed above), in most weeks preceding or following an observed peak, a proportion of the forecast trajectories were positively correlated with the pre-peak and post-peak case counts (Figure I in S1 Text). This indicates that the forecasts could at least partially characterise the qualitative trends in future case counts (i.e., whether case counts would increase, decrease, or remain stable).

## Relative model performance

Recall that at each week of the study period, sample trajectories of daily case counts were provided over four-week horizons for each Australian jurisdiction. For each individual daily case count prediction, we ranked the individual models and

the ensemble using CRPS on log-transformed values [38]. The results for days where at least one case was reported are shown in Fig 6. Collectively, the individual models were more likely than the ensemble to be the top-ranked forecast, but no single model dominated the top ranking. The strength of the ensemble was that it was *most often the best or second-best* forecast. Skill scores for each model, relative to the ensemble, were consistently below zero, further reinforcing the finding that no model consistently out-performed the ensemble (Fig 7). The model rankings for each ensemble forecast (Figure J in S1 Text) also demonstrate that there were no obvious predictors of which model would perform the best for a given time period or epidemiological context.

### Pre-Delta: May to October 2020

We now highlight how the forecasts supported public health activities in each phase of the study period, beginning with the Pre-Delta phase. This first period of sustained local COVID-19 transmission began in mid-2020, where the majority of cases occurred in the state of Victoria. Ensemble forecasts for this period are presented in Figure A in S1 Text. The performance of the UoM forecasts over this second wave has already been reported [12], and we begin by comparing the ensemble forecasts to the UoM forecasts over this wave.

In the weeks prior to the peak in daily cases (3 August 2020), the upper bounds of the UoM forecasts substantially overshot the true peak, and predicted sustained exponential growth in cases. This exponential growth was tempered in the ensemble, which averaged over three models (MU, UoA, and UoM). Accordingly, the ensemble outperformed the UoM forecasts (which had a mean skill score of -0.37, see Figure K in S1 Text), even though the ensemble forecasts tended to undershoot the data. By late July, we identified that case ascertainment and reporting delays had substantially increased,

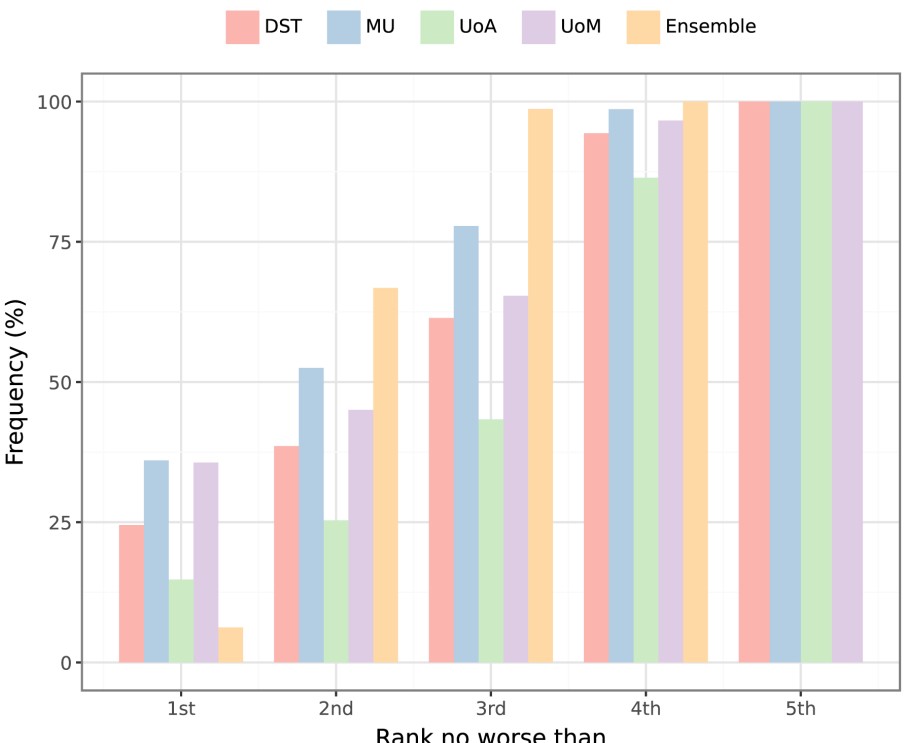

**Fig 6. Cumulative rankings for each individual model and for the ensemble.** Rankings are shown for days where at least one case was reported, calculated using CRPS on log-transformed values. While the ensemble was rarely the top-ranked model, it was *at least* 2nd-best approximately two-thirds of the time (66.7%), and was *almost always* in the top three (98.7%).

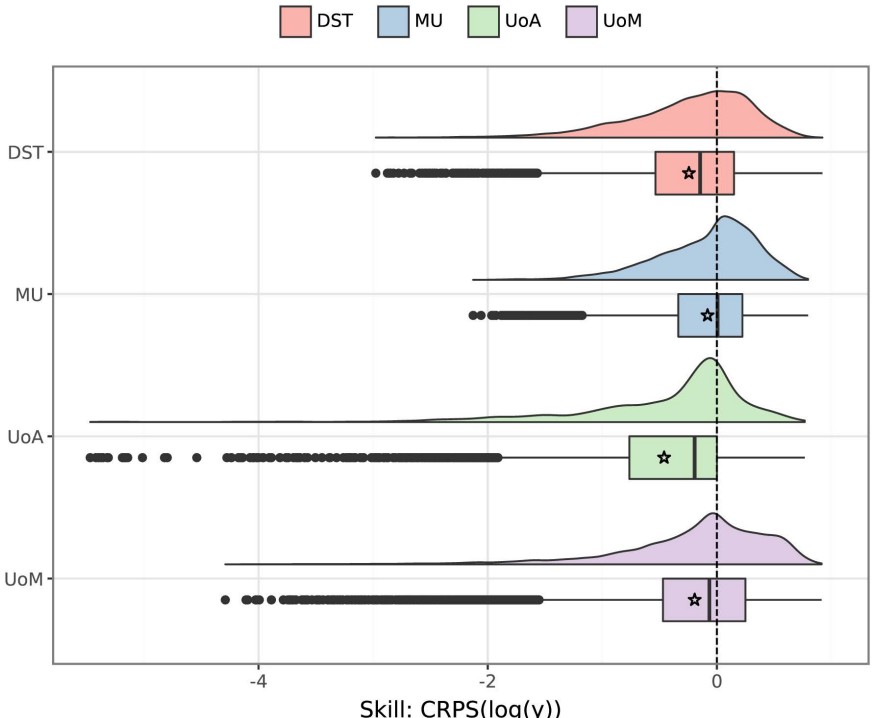

**Fig 7. Distribution of skill scores for each individual model, measured against the ensemble.** Results are shown for days where at least one case was reported (around 53% of all observations). Skill scores were calculated using CRPS on log-transformed values [38]. Positive values indicate performing better than the ensemble (with a maximum skill of 1 for a perfect point forecast), negative values indicate performing worse than the ensemble, and the vertical dashed line marks the threshold skill of zero (performance equal to the ensemble). Stars indicate mean skill scores. The MU model performed better than the other models, with a median skill score of 0.002. Mechanistic models performed well for extended periods of zero cases (median skill scores of 0.75–0.97) but were otherwise outperformed by the ensemble, with median scores of −0.07 (UoM), −0.14 (DST), and −0.19 (UoA).

and our right-truncation adjustments were under-estimating the true case counts in the most recent days prior to each forecast. In the weeks after the peak, the UoM forecasts confidently predicted a sustained decrease in cases, while the MU and UoA models exhibited broader credible intervals and tended to overshoot the data. As a result, the UoM forecasts performed better than the ensemble forecasts (mean skill score of 0.31, see Figure K in S1 Text).

Daily case incidence in Victoria steadily decreased from the August peak. In early September the Victorian government announced a gradual easing of restrictions, subject to reaching specific 14-day moving average case thresholds. Because the ensemble forecast comprised individual trajectories from each contributing model, we were able to calculate the proportion of all trajectories that satisfied a given 14-day case threshold on each date in the forecast horizon, and reported this as the daily probability of achieving each target threshold. These predictions were provided to the Victorian government throughout September and October (and are shown in Moss et al. [12]).

## Delta: June to December 2021

The onset of the Delta variant in June 2021 marked the first instance of sustained local COVID-19 transmission for most Australian jurisdictions, and public health responses focused on preventing local transmission. Accordingly, daily case counts remained very low in most jurisdictions, only exceeding 25 cases per day in the Australian Capital Territory (peak of 51), New South Wales (peak of 1,495), and Victoria (peak of 1,955). Ensemble forecasts for this period are presented in Figure B in S1 Text.

Perhaps the single greatest value of the forecasts in this period was to demonstrate how rapidly case counts could increase if local transmission was left unchecked.

In May 2021, New South Wales Health set a target of achieving 80% coverage of the adult population with two vaccination doses by the end of December 2021 [42]. This was followed in August 2021 by an interim target of 70% coverage of the adult population with two doses, to encourage vaccine uptake and to begin easing restrictions for fully vaccinated people [42]. As a consequence, there was substantial vaccine roll-out over the 4-week forecast horizons from August onward. This was accompanied by rapid model development from the DST and UoM teams, with the effects of vaccination incorporated into these models in September 2021. The statistical MU model did not require explicit adjustment for vaccination effects, although vaccination was indirectly accounted for in the model's structure and updated parameter estimates based on post-vaccine global case counts.

The forecasts were generally in very good agreement with the data in each jurisdiction. In particular, the mean CRPS values for New South Wales and Victoria were consistently lower than for the Pre-Delta wave in Victoria (see Figure G in S1 Text).

## Omicron BA.1: January to March 2022

This wave coincided with a national pivot from pursuing strong suppression to reopening and management of substantial levels of local COVID-19 transmission. Ensemble forecasts for this period are presented in Figure C in S1 Text.

The forecasts struggled to predict the massive surge and subsequent decline in daily case counts, as demonstrated by decreased coverage (Fig 4), increased bias (Fig 5, Figures L and M in S1 Text), and worse coverage than other waves (Figures N and O in S1 Text).

This was due, at least in part, to mechanistic models failing to account for significant reduction in vaccine protection against Omicron, rapid changes in case ascertainment [35], and reduced mixing due to school holiday effects over December and January. Also in January, the vaccine roll-out began for children aged 5–11 years, and booster doses were recommended for at-risk individuals [43].

A primary concern in all jurisdictions was the timing of the epidemic peak, because the substantial increase in cases was impacting workforce capacity and causing disruptions to food supply chains, which in turn prompted panic buying [44]. The forecasts consistently predicted that the peaks would occur around 2 weeks later than they actually occurred (and for the mechanistic models, this was robust to adjustments such as relaxing assumptions regarding case ascertainment and immunity). Despite this inaccuracy, the forecast predictions that case activity would peak and begin to decrease in a matter of weeks remained an important and useful message for government.

## Omicron BA.2/BA.5: April to August 2022

This period saw the gradual replacement of Omicron BA.1 with Omicron BA.2 and BA.5, and case counts in all jurisdictions steadily decreased after the large Omicron BA.1 peaks. Ensemble forecasts for this period are presented in Figure D in S1 Text.

The forecasts reported on 11 June, generated from daily case counts up to 31 May (inclusive), predicted downwards trends in all jurisdictions, consistent with the trends in the most recent case counts. On 16 June we received additional genomic data from New South Wales that allowed us to estimate the transmission advantage of Omicron BA.5, relative to Omicron BA.1 and BA.2. By incorporating this transmission advantage into the mechanistic models, our next forecasts predicted upwards trends in all jurisdictions *despite no such trend in the reported aggregate data*. Consistent with these predictions, daily case counts began to increase later in June, and all jurisdictions experienced a peak in July (see Figure D in S1 Text).

By predicting this inflection point before it occurred, and by doing so based on appropriate interpretation of relevant local data, these forecasts helped convince public health stakeholders that the current downwards trend in cases would not be sustained.

**Multiple variants: September to December 2022**

Following the peak and decline of Omicron BA.5, Australia experienced circulation of numerous COVID-19 variants, with no single variant becoming dominant. Data from international contexts provided evidence that many of these variants had growth advantages over Omicron BA.1, BA.2, and BA5, although it was unclear how these estimates might translate to Australia's immune landscape. Ensemble forecasts for this period are presented in Figure E in S1 Text.

Relative to the preceding Omicron BA.2/BA.5 period, the forecasts exhibited similarly good coverage (Fig 4) and performance (see Figure P in S1 Text). When case counts began to increase in November, the forecasts were confident that the epidemic peaks would be similar in size, or smaller than, the Omicron BA.5 peaks. This was a reassuring message that would prove to be accurate, and brings us to the end of our study period.

**Communication of outputs**

Over the 3.5 years of work summarised here, an ongoing concern was to ensure that model predictions and other quantitative outputs were reported in such a way that our stakeholders (spanning state, territory, and national governments and public health committees) would be able to interpret them appropriately and accurately act on and communicate their implications. While we did not have the capacity to undertake formal assessments of how the 152 weekly situation reports that we produced over this study period were interpreted, as described by McCaw and Plank [45] we did have open communication channels with our stakeholders and were able to explore and refine our reporting and visualisations in an iterative manner.

Fig 8 shows several examples of visualisations included in these reports. For example, our primary form of communicating the forecasts was daily credible intervals (see, e.g., Fig 3). However, this form can obscure the predicted size and timing of an epidemic peak, and so we explored the use of density plots to extract these features from individual forecast trajectories (Fig 8A). Another simple, yet surprisingly useful, modification to the daily credible interval figures was to overlay several randomly-selected forecast trajectories (Fig 8B), which helped to avoid the credible interval contours being misinterpreted as case count trajectories. Finally, plotting past forecasts against the most recent data (Fig 8C) was a simple way of communicating recent forecast performance that provided stakeholders with a qualitative basis for deciding how much trust to place in the current forecasts.

## Discussion

### Principal findings

Similar to findings reported in COVID-19 forecasting studies in other countries [6–11,41], the use of an ensemble to combine forecasts from multiple models improved forecast performance and reliability (Figs 4–6). Our ensemble forecasts provided information that supported public health responses throughout the study period, from the early periods of preventing local transmission, to the later periods of sustained local transmission and concerns about the burden placed on healthcare systems. Notable outcomes during the study period include:

- Pre-Delta: we used the forecasts to predict when 14-day average case targets would be achieved, which would trigger the gradual easing of mobility restrictions in the state of Victoria.

- Delta: this was the first instance of sustained local transmission for most jurisdictions, and the forecasts were useful for illustrating how rapidly case counts could increase if transmission was not curtailed.

- Omicron BA.1: the peak occurred 1–2 weeks earlier than the forecasts predicted, but forecast predictions that activity would peak and then decline in a matter of weeks was reassuring in the face of workplace absenteeism that disrupted supply chains.

 

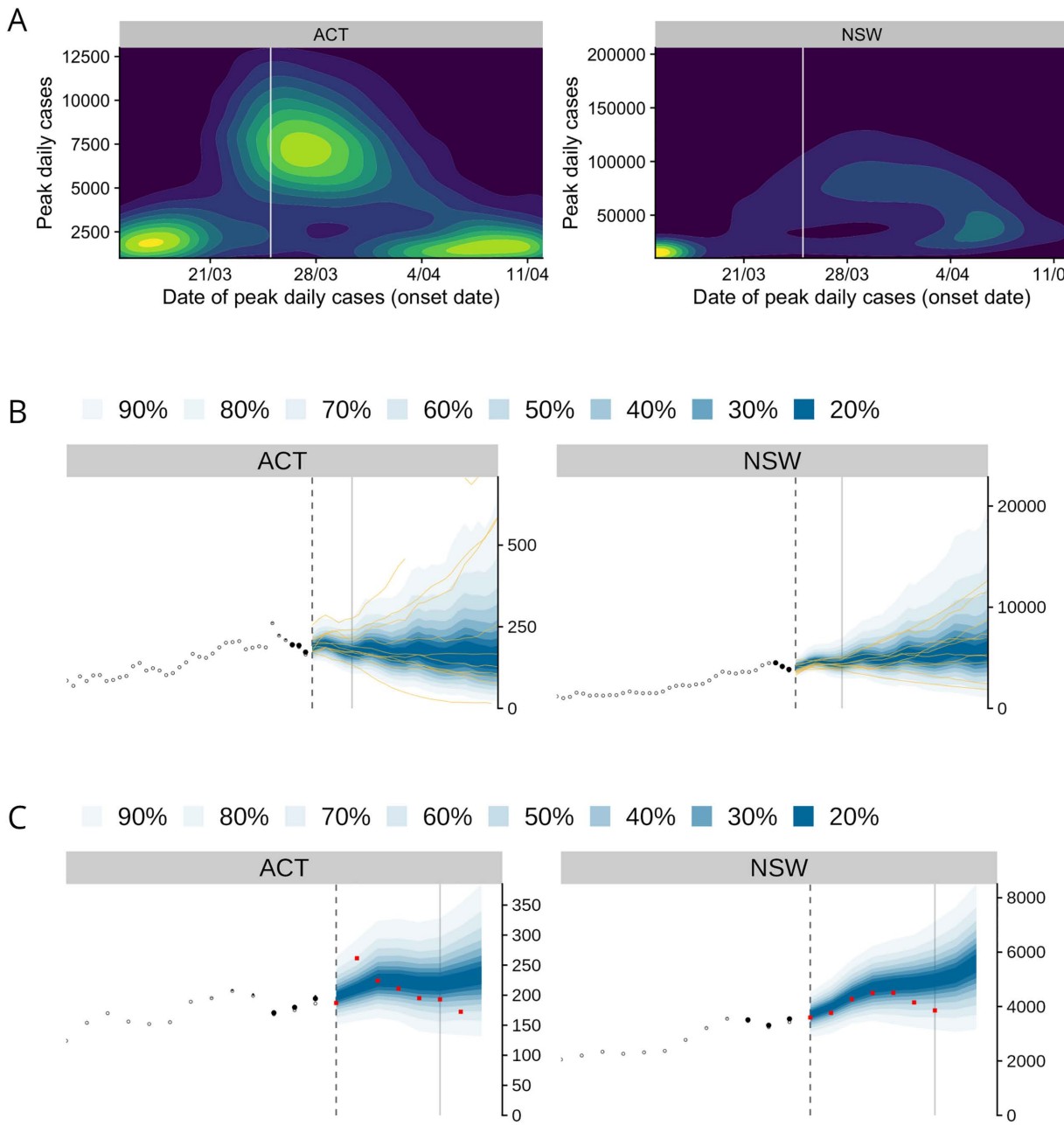

**Fig 8. Example figures from weekly situation reports.** A: Density plots showing predicted peak timing and peak case incidence, from the report produced on 24 March 2022. B: Ensemble forecasts of new daily local cases, showing credible intervals and sample trajectories, from the report produced on 25 November 2022. C: Retrospective evaluation of ensemble forecasts from the previous week against new data (red points), from the report produced on 25 November 2022.

- Omicron BA.2/BA5: the forecasts were able to correctly predict an increase in cases while the current case counts were decreasing, and this was a particularly useful message for health protection committees.

- Multiple variants: the forecasts accurately predicted that the peaks in late 2022 would not be larger than the Omicron BA.5 peaks.

PLOS Computational Biology

## Study strengths

The ensemble approach allowed us to readily incorporate forecasts contributed by teams across Australia, and provided a framework for rapid evaluation of new model iterations for potential inclusion in the ensemble. The weekly performance evaluations and model inclusion/exclusion decisions relied on measures of forecast error (using CRPS) under the assumption that existing policies would persist for the entire forecast horizon. However, as we have described above, the ensemble forecasts influenced policy decisions throughout the study period, and some of these decisions directly impacted local COVID-19 transmission and/or case ascertainment. Accordingly, when policy decisions were understood to have influenced the case data reported over a forecast horizon, we factored these effects into our analyses. In particular, where the difference between forecasts and reported data were consistent with the likely effects of the policy decisions, we considered this a successful outcome [46].

We deliberately chose to use equal-weight ensemble forecasts and, as we explain in the methods section, each model was evaluated against the most recent data rather than competing against the other models for inclusion in the ensemble [37]. As we have discussed in the results section, a model's ranking in recent weeks was not a reliable predictor of that model's ranking in future weeks. The strongest finding we have regarding the rankings is that the ensemble was either the best, or second-best, more often than any individual model. Both findings are consistent with ensemble COVID-19 forecast evaluations in other countries [9,10].

Finally, we note here that the ensemble forecasts presented here also served as an input for COVID-19 hospital bed occupancy forecasts (December 2021 to December 2023) that further supported Australia's public health responses [19].

## Study limitations

While pursuing the prevention of local transmission, Australian jurisdictions maintained high testing levels and the proportion of infected persons that were identified as cases was likely to be both very high, and to remain relatively constant. The ensemble forecasts performed very well during this period (the "Pre-Delta" and "Delta" phases).

When Australia transitioned to re-opening, case ascertainment was substantially reduced, and was challenging to estimate in near-real-time [35]. By February 2022, more than 94% of people over the age of 16 were fully vaccinated [47], and while the mechanistic models were subject to numerous adaptations and refinements to account for vaccination coverage, booster vaccinations, and waning immunity, throughout 2022 it became more challenging to estimate population susceptibility to the emerging variants of concern. This was due to an increasingly complex population immune landscape, resulting from heterogeneous levels of vaccination and immunising exposure, and challenges in estimating the immunogenicity of emerging variants.

However, despite these challenges, the inclusion of both mechanistic and statistical models in the ensemble substantially improved the forecast performance and reliability across the entire study period. The only period where the forecast did not capture the case data was the rapid emergence and decline of Omicron BA.1, which occurred as Australia pivoted from pursuing strong suppression to reopening, accepting substantial levels of local transmission, and rapid antigen tests were approved for self-testing and reporting [48].

Two of the three teams that participated during the "Omicron BA.1" period (DST and UoM) used mechanistic models that explicitly incorporated case ascertainment and immunity dynamics. These teams explored adjustments to case ascertainment and immunity assumptions, on the grounds that this might result in smaller and earlier predicted peaks, closer to the observed peaks. However, both teams reported that when these adjusted models were fit to the reported (pre-peak) case data, estimates for other parameters also changed in response to these adjustments, and this resulted in similar forecasts to the original models (i.e., peaks were larger and later than observed).

## Conclusion

Our ensemble COVID-19 forecasts were produced under contract with the Australian Government Department of Health and Aged Care, and participation was limited to a small number of institutes. In contrast, the United States of America and Europe both established public COVID-19 forecast hubs that had open submission policies and included all submissions

that complied with the hubs' technical requirements in their ensemble forecasts [9,10]. This open nature resulted in large numbers of participating teams, with 67 teams contributing US forecasts (July 2020 to December 2021 [10]), and 48 teams contributing European forecasts (March 2021 to March 2022 [9]).

Including great numbers of models in an ensemble is generally understood to improve the forecast performance, with diminishing improvements once a sufficient number of models is reached [49]. In the context of influenza-like illness (ILI) and COVID-19 forecasts in the United States of America, a recent study concluded that hubs should target "a minimum of four validated forecast models to ensure robust performance" [50]. In contrast to the USA and European hubs, our ensemble included an average of 3.03 models (see Table 3) and the interval where all four teams contributed forecasts occurred during the Omicron BA.1 wave (see Fig 1) where the ensemble performance was lowest, and so we cannot draw any conclusions about whether our ensemble forecast's performance may have been influenced by how many of the teams were included in the ensemble at any given week.

Forecast evaluations from both hubs reported findings that are broadly consistent with those reported here. Forecasts performed well in periods of stable behaviour, but struggled at longer horizons around inflection points, and individual models varied widely in their ability to account for new COVID-19 variants. The ensemble forecasts were consistently among the best-performing forecasts across all horizons and locations, and weighting individual models by their past performance did not improve the ensemble performance. The US hub reported that forecast prediction intervals were generally over-confident and had low coverage, particularly when case numbers were changing rapidly [10], similar to our findings for the Omicron BA.1 wave (Fig 4). Both hubs also reported that their open submission policies limited the possibility to understand drivers of forecast performance, with many teams participating at different times, participating intermittently, and providing varied and/or limited descriptions of their methods [9,10].

The ensemble forecasts presented here played an important role in supporting Australian public health decision-making over a wide range of epidemiological and policy contexts. Consistent with reflections from modelling and data analysis communities around the world [45,51–53], frequent communication between modellers and public health stakeholders — and the mutual understanding and trust that this fostered — was integral to the utility of these ensemble forecasts. Being organised as a consortium, with several members sitting on key public health committees, meant that there were direct lines of communication between decision makers and modellers, and this in turn expedited the prototyping and development of effective analyses and communications.

More broadly, the collective role of our consortium as a means for weekly peer-review of model forecasts, and the weekly decision of which model iteration(s) to include, directly supported rapid model development and evaluation, while also ensuring that only "known good" model iterations were contributing to the ensemble forecast. Having common performance targets and evaluation processes for all models, conducted openly within the consortium each week, fostered mutual collaboration. This collaborative structure also helped to avoid duplication of effort, which was extremely beneficial at times of high stress, urgent delivery schedules, and unsustainable workloads. Such an approach may be challenging to adapt to the scale of the US and European forecast hubs, but given the large number of teams that contributed to each of those hubs, the ensemble forecasts produced by the hubs would not have been substantially influenced by the inclusion or exclusion of any single model. In contrast, given the small number of teams that contributed to our ensemble, the inclusion or exclusion of a single model could exert much larger influence on the ensemble forecasts.

One benefit of this organisation, and the small size of our consortium relative to similar groups overseas, was retaining the human element in selecting which model iteration(s) to include in the ensemble, rather than using an arbitrary quantitative threshold or simply including all available forecasts in the ensemble. This allowed us to explain features of each ensemble forecast, and the underlying rationale for these features in terms of the contributing models and their assumptions, which helped our public health stakeholders to interpret the forecasts appropriately.

## Supporting information

**S1 Text. Technical appendix.** Contains supporting figures.
(PDF)

## Acknowledgments

National Notifiable Disease Surveillance System data on COVID-19 were provided by the Office of Health Protection and Response, Australian Government Department of Health and Aged Care, on behalf of the Communicable Diseases Network Australia (CDNA). We thank public health staff from incident emergency operations centres in state and territory health departments, and the Australian Government Department of Health and Aged Care, along with state and territory public health laboratories. We thank members of CDNA for their feedback and perspectives on the study results. This research was supported by use of the Nectar Research Cloud, a collaborative Australian research platform supported by the NCRIS-funded Australian Research Data Commons (ARDC), and by the University of Melbourne Research Computing Services.

## Author contributions

**Conceptualization:** Rob J Hyndman, David J Price, James M McCaw, Freya M Shearer.

**Data curation:** Gerard E Ryan, Tianxiao Hao, Aarathy Babu, Katharine L Senior, David J Price.

**Investigation:** Robert Moss, Ruarai J Tobin, Mitchell O'Hara-Wild, Adeshina I Adekunle, Dennis Liu, Tobin South, Dylan J Morris, Gerard E Ryan, Tianxiao Hao, Aarathy Babu, Katharine L Senior, James G Wood, Nick Golding, Joshua V Ross, Peter Dawson, Rob J Hyndman, David J Price, James M McCaw, Freya M Shearer.

**Methodology:** Robert Moss, Ruarai J Tobin, Mitchell O'Hara-Wild, Adeshina I Adekunle, Dennis Liu, Tobin South, Dylan J Morris, Gerard E Ryan, Tianxiao Hao, Aarathy Babu, Katharine L Senior, James G Wood, Nick Golding, Joshua V Ross, Peter Dawson, Rob J Hyndman, David J Price, James M McCaw, Freya M Shearer.

**Supervision:** David J Price, James M McCaw, Freya M Shearer.

**Visualization:** Robert Moss, Ruarai J Tobin, Mitchell O'Hara-Wild, Gerard E Ryan, David J Price, James M McCaw, Freya M Shearer.

**Writing – original draft:** Robert Moss.

**Writing – review & editing:** Robert Moss, Ruarai J Tobin, Mitchell O'Hara-Wild, Adeshina I Adekunle, Dennis Liu, Tobin South, Dylan J Morris, Gerard E Ryan, Tianxiao Hao, Aarathy Babu, Katharine L Senior, James G Wood, Nick Golding, Joshua V Ross, Peter Dawson, Rob J Hyndman, David J Price, James M McCaw, Freya M Shearer.

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
