## [Decision Letter · Decision Letter 0]

2 Dec 2025

Ensemble forecasts of COVID-19 activity to support Australia's pandemic response: 2020–22

PLOS Computational Biology

Dear Dr. Moss,

Thank you for submitting your manuscript to PLOS Computational Biology. After careful consideration, we feel that it has merit but does not fully meet PLOS Computational Biology's publication criteria as it currently stands. Therefore, we invite you to submit a revised version of the manuscript that addresses the points raised during the review process.

We look forward to receiving your revised manuscript.

Kind regards,

Guillermo Lorenzo

Academic Editor

PLOS Computational Biology

Thomas Leitner

Section Editor

PLOS Computational Biology

**Journal Requirements:**

At this stage, the following Authors/Authors require contributions: Robert Moss, Ruarai J Tobin, Mitchell O'Hara-Wild, Adeshina I Adekunle, Dennis Liu, Tobin South, Dylan J Morris, Gerard E Ryan, Tianxiao Hao, Aarathy Babu, Katharine L Senior, James G Wood, Nick Golding, Joshua V Ross, Peter Dawson, Rob J Hyndman, David J Price, James M McCaw, and Freya M Shearer. Please ensure that the full contributions of each author are acknowledged in the "Add/Edit/Remove Authors" section of our submission form.

5) We have noticed that you have uploaded Supporting Information files, but you have not included a list of legends. Please add a full list of legends for your Supporting Information files after the references list.

State the initials, alongside each funding source, of each author to receive each grant. For example: "This work was supported by the National Institutes of Health (####### to AM; ###### to CJ) and the National Science Foundation (###### to AM).".

**Reviewers' comments:**

Reviewer's Responses to Questions

**Comments to the Authors:**

Reviewer #1: See attached

Reviewer #2: Firstly, I would like to thank the authors for their service and work throughout the pandemic. Secondly, I would like to thank them for the submission that reflects on their results and the impact on public policy in Australia – I thoroughly enjoyed reading the manuscript and found it refreshing that they are willing to reflect on their work throughout the pandemic. This manuscript focusses on the results and reflects of an ensemble method using for forecasting daily COVID-19 cases in Australia, combining models from 4 different modelling groups who were asked to contribute towards pandemic efforts, and describes the consortium’s internal process for the exposition of the results for a wider audience and its successes. In particular, the manuscript demonstrated some results of the ensemble model and showed some specific case studies where the approach performed well (and often better than the individual models), then went into a discussion on the findings, strengths and limitations, and the implications. Whilst I enjoyed reading the manuscript and its reflections, it does not seem to be presenting any novel mathematics or even a novel application of the mathematics, it seems to be only novel as a geographic location. It is a good account of the history of the consortium and the modelling efforts of the time, but that seems to be it? Whilst that, in itself, is not enough reason to reject the manuscript (there are plenty of examples in PLoS Comp Biol that are doing this), I think the manuscript needs a bit more care - the manuscript reads like a large situational report, rather than an academic journal article.

My recommendation for this submission is Major Revision.

The major issues I have are as follows:

1. The introduction needs to have a more substantial literature review of ensemble models, where they have been used and to what effect.

2. The introduction needs to highlight explicitly what is novel about this work, and clearly highlight what the take home message is about the approach conducted.

3. The method section should outline more information about how the ensemble model is actually constructed (there are random bits of information throughout the manuscript, such as equal weighting in Section 4.2).

4. There seems to be something interesting that the authors make a large point about: “the models were evaluated against the most recent data rather than competing against the other models …, with the purpose of deciding whether the model outputs were likely to be a reliable basis for decision making.” It is not clear from the text why this is relevant and important?

5. There should be a discussion about when one should produce an ensemble model and when one should not.

6. Section 4.4: a lot of this implies that having a small ensemble is better, which I don’t think is necessarily true in general and could be quite misleading for readers. It would be better if the authors also included the positives that came out of the other two hubs, so the reader can make an informed decision themselves.

The minor issues I have are as follows:

1. How and why did the authors impute missing symptom onset dates? Imputing in general is widely criticised (e.g., DOI: 10.1177/2192568218811922, DOI: 10.1371/journal.pone.0111964)

2. Figure 1: it might be good to highlight more substantial changes in the pandemic curve here (e.g., vaccination). Can the authors also clarify that it is correct that there were less than 10 cases from late December 2020 to approximately June 2020? If that is true, I would suggest writing something in the text as this was not expected (and I suspect would not be expected by readers outside of Australia).

3. In the formula for forecast bias, what are Y_t and y_t? The manuscript refers to bias quite a bit so it would be appropriate to include more information about it. Similarly for CRPS, the authors should include more details

4. Figure 3: why do the credible regions look so disjoint? I think it is due to the refreshing of model outputs? If so, please make this clearer. I would also suggest making the y-axis the same for either all the figures or for the ones the authors want the reader to compare (probably the same y-axis for each of the regions)

5. Table 4: in the caption I would write out CRPS again. I am also struggling to interpret Table 4 without the raw CRPS values. It also seems, from the text, that the improvement is from comparing to the “4 weeks”? Can the authors make this clearer in the caption please.

6. On Line 186, why do forecasts improve for “Omicron BA.2/BA.5” and “Multiple variants”?

7. Figure 7: how is the score for each model measured against the ensemble? What does the dashed line represent? I don’t see how this Figure shows that no model consistently outperformed the ensemble.

8. As much as I enjoyed reading Section 3.9, I don’t see what it actually brings to the manuscript? (Particularly since a lot of it can be attributed to [35])

9. Line 415: this reads like the authors were also trying to predict government policy and its impact, and that the ensemble failed to do so – this failure should be completed expected since the models would not be able to do this either and it would be unreasonable to expect them (and the ensemble) to do so. I think the sentence 413 to 415 is not needed at all and negatively changes the narrative.

Reviewer #3: Overall, this was a very interesting summary of the impactful work conducted in Australia during the COVID-19 pandemic. I think it would be of interest to the PLOS CB audience. At times, the paper felt like more of a report than a research article, and I encourage the authors to consider adding some more mathematical context for the models mentioned.

- In some places, it isn't clear that the consortium reporting to the government included the authors of this work. I encourage the authors to go back through the abstract and introduction and make sure it is clear that they were a part of this consortium. Alternatively, if they were not, then I would make this distinction clearer.

- Could some more explicit details be given for the reader around the models, and data integration. I understand this might be challenging given the evolving nature of the models and the amount of data used, but I think it would be helpful for the reader to understand the types of models better so that we can understand their comparison.

**Have the authors made all data and (if applicable) computational code underlying the findings in their manuscript fully available?**

The PLOS Data policy requires authors to make all data and code underlying the findings described in their manuscript fully available without restriction, with rare exception (please refer to the Data Availability Statement in the manuscript PDF file). The data and code should be provided as part of the manuscript or its supporting information, or deposited to a public repository. For example, in addition to summary statistics, the data points behind means, medians and variance measures should be available. If there are restrictions on publicly sharing data or code —e.g. participant privacy or use of data from a third party—those must be specified.requires authors to make all data and code underlying the findings described in their manuscript fully available without restriction, with rare exception (please refer to the Data Availability Statement in the manuscript PDF file). The data and code should be provided as part of the manuscript or its supporting information, or deposited to a public repository. For example, in addition to summary statistics, the data points behind means, medians and variance measures should be available. If there are restrictions on publicly sharing data or code —e.g. participant privacy or use of data from a third party—those must be specified.requires authors to make all data and code underlying the findings described in their manuscript fully available without restriction, with rare exception (please refer to the Data Availability Statement in the manuscript PDF file). The data and code should be provided as part of the manuscript or its supporting information, or deposited to a public repository. For example, in addition to summary statistics, the data points behind means, medians and variance measures should be available. If there are restrictions on publicly sharing data or code —e.g. participant privacy or use of data from a third party—those must be specified.requires authors to make all data and code underlying the findings described in their manuscript fully available without restriction, with rare exception (please refer to the Data Availability Statement in the manuscript PDF file). The data and code should be provided as part of the manuscript or its supporting information, or deposited to a public repository. For example, in addition to summary statistics, the data points behind means, medians and variance measures should be available. If there are restrictions on publicly sharing data or code —e.g. participant privacy or use of data from a third party—those must be specified.

Reviewer #1: Yes

Reviewer #2: Yes

Reviewer #3: Yes

PLOS authors have the option to publish the peer review history of their article (what does this mean?). If published, this will include your full peer review and any attached files.). If published, this will include your full peer review and any attached files.). If published, this will include your full peer review and any attached files.). If published, this will include your full peer review and any attached files.

...

Reviewer #1: No

Reviewer #2: No

Reviewer #3: No

**Figure resubmission:**

**Reproducibility:**



---

## [Decision Letter · Decision Letter 1]

6 Apr 2026

Dear Dr. Moss,

We are pleased to inform you that your manuscript 'Ensemble forecasts of COVID-19 activity to support Australia's pandemic response: 2020–22' has been provisionally accepted for publication in PLOS Computational Biology.

Best regards,

Guillermo Lorenzo

Academic Editor

PLOS Computational Biology

Thomas Leitner

Section Editor

PLOS Computational Biology

Reviewer's Responses to Questions

**Comments to the Authors:**

Reviewer #1: The authors have done a very thorough job addressing all of this reviewer's concerns, and this reviewer is now happy to recommend the article for acceptance. The authors deserve particular commendation for their extensive work in revising the mathematical rigor and clarity of the article, as well as their thoughtful responses in their letter.

This reviewer does have a few minor comments, which the author and/or editor are free to ignore, as they reflect personal stylistic preference.

1. The use of t and T as indexes for each modeling team might potentially confuse, as t is near-universally used as a time index. Note there is no ambiguity in the paper itself; the authors are consistent and do not muddle the meaning of t. Nonetheless, perhaps g and G (for "group") may avoid this issue.

2. Is the vertical orientation of Figure 7 optimal? This reviewer rotated the image to read it horizontally, and found that orientation clearer.

3. It might be better to move figure 8 to a supplementary material, as its purpose is not in the findings of the current article as such, but in an (interesting) related discussion concerning communication of outputs to stakeholders. This reviewer recommends keeping the first paragraph of 3.9, and other information in an appendix. By the way, the authors may also consider writing a commentary manuscript discussing such communication aspects in a public health journal that publishes this sort of work - such a contribution would be useful.

This reviewer would like to once again thank the authors for their work.

Reviewer #2: I would like to thank the authors for their attention to (all) the reviewer comments. I am satisfied with their changes, this reads significantly more like a research article now and I am happy to accept this version. There are some minor typos throughout though, so please double check everything.

Reviewer #3: I thank the authors for responding to my comments

**Have the authors made all data and (if applicable) computational code underlying the findings in their manuscript fully available?**

The PLOS Data policy requires authors to make all data and code underlying the findings described in their manuscript fully available without restriction, with rare exception (please refer to the Data Availability Statement in the manuscript PDF file). The data and code should be provided as part of the manuscript or its supporting information, or deposited to a public repository. For example, in addition to summary statistics, the data points behind means, medians and variance measures should be available. If there are restrictions on publicly sharing data or code —e.g. participant privacy or use of data from a third party—those must be specified.requires authors to make all data and code underlying the findings described in their manuscript fully available without restriction, with rare exception (please refer to the Data Availability Statement in the manuscript PDF file). The data and code should be provided as part of the manuscript or its supporting information, or deposited to a public repository. For example, in addition to summary statistics, the data points behind means, medians and variance measures should be available. If there are restrictions on publicly sharing data or code —e.g. participant privacy or use of data from a third party—those must be specified.requires authors to make all data and code underlying the findings described in their manuscript fully available without restriction, with rare exception (please refer to the Data Availability Statement in the manuscript PDF file). The data and code should be provided as part of the manuscript or its supporting information, or deposited to a public repository. For example, in addition to summary statistics, the data points behind means, medians and variance measures should be available. If there are restrictions on publicly sharing data or code —e.g. participant privacy or use of data from a third party—those must be specified.requires authors to make all data and code underlying the findings described in their manuscript fully available without restriction, with rare exception (please refer to the Data Availability Statement in the manuscript PDF file). The data and code should be provided as part of the manuscript or its supporting information, or deposited to a public repository. For example, in addition to summary statistics, the data points behind means, medians and variance measures should be available. If there are restrictions on publicly sharing data or code —e.g. participant privacy or use of data from a third party—those must be specified.

Reviewer #1: Yes

Reviewer #2: Yes

Reviewer #3: None

PLOS authors have the option to publish the peer review history of their article (what does this mean?). If published, this will include your full peer review and any attached files.). If published, this will include your full peer review and any attached files.). If published, this will include your full peer review and any attached files.). If published, this will include your full peer review and any attached files.

...

Reviewer #1: No

Reviewer #2: No

Reviewer #3: No

---

## [Editor Report · Acceptance letter]

PCOMPBIOL-D-25-01870R1

Ensemble forecasts of COVID-19 activity to support Australia's pandemic response: 2020–22

Dear Dr Moss,

I am pleased to inform you that your manuscript has been formally accepted for publication in PLOS Computational Biology. Your manuscript is now with our production department and you will be notified of the publication date in due course.

With kind regards,

Lilla Horvath
